# Marital Status of Never Married with Rey Auditory Verbal Learning Test Cognition Performance Is Associated with Mild Cognitive Impairment

Yohn Jairo Parra Bautista [1,*], Samia S. Messeha [2,3], Carlos Theran [1], Richard Aló [1], Clement Yedjou [3], Victor Adankai [1], Samuel Babatunde [1] and on behalf of the Alzheimer's Disease Prediction of Longitudinal Evolution (TADPOLE) [†]

1 Department of Computer Information Sciences, College of Science and Technology, Florida A&M University, Tallahassee, FL 32307, USA

2 College of Pharmacy and Pharmaceutical Sciences, College of Science and Technology, Florida A&M University, Tallahassee, FL 32307, USA

3 Department of Biological Sciences, College of Science and Technology, Florida A&M University, Tallahassee, FL 32307, USA

* Correspondence: yohn.parrabautista@famu.edu

† Secondary data used in preparation of this article were obtained from the Alzheimer's Disease Prediction of Longitudinal Evolution(TADPOLE) database (https://tadpole.grand-challenge.org/Data/). As, such, the researchers within the TADPOLE contributed to the design and implementation of TADPOLE and/or provided data but did not participate in analysis or writing of this prediction challenge. A complete listing of TADPOLE key contributors can be found at: https://tadpole.grand-challenge.org/Contact/.

**Abstract:** A small number of robust studies have explored the association between cognitive tests and marital status levels of mild cognitive impairment (MCI) patients using the TADPOLE dataset. Rey Auditory Verbal Learning Test (RAVLT) cognition performance combined with marital status levels is associated with increased odds of MCI than either RAVLT in isolation. The cross-sectional association between RAVLT performance in immediate response, learning, forgetting, and perception of forgetting with marital status and MCI was evaluated using TADPOLE data. We included participants with MCI and normal cognition in our study. Based on our logistic regression model, four RAVLT subgroups are associated with MCI (low and high response performance, immediate response with learning, immediate response with learning and forgetting, immediate response with learning, forgetting, and perception of forgetting). We adjusted models for sex, age, race, marital status, education, ethnicity, APOE4 genotype, hippocampus, whole brain, ventricles, and ICV. A mean age of 77/67 years was observed in the sample (n = 6560), 44% of participants were females, and 58% had mild cognitive impairment. Subgroups whose ages are 61 to 70 (OR 0.26, 95% CI 0.15–0.45) and older (OR 0.07, 95% CI 0.04–0.12), as well as race: black/African American (OR 0.13, 95% CI 0.03–0.52), multiple races (OR 0.05, 95% CI 0.01–0.24), and never married (OR 0.2, 95% CI 0.12–0.34) were negatively associated with immediate response and forgetting subgroup tests. There is a need for studies that evaluate other cognitive tests in the TADPOLE dataset with missing data as a predictive tool that aligns with the factors associated with MCI.

**Keywords:** cognitive test; education; cognitive impairment; dementia

## 1. Introduction

A complex neurological disorder, Alzheimer's disease (AD) severely affects memory and cognitive function in millions of people worldwide, leading to dementia [1]. A neuropathological diagnosis of AD is usually made in patients who exhibit criteria for dementia [2]. In mild cognitive impairment (MCI), the preclinical transitional stage between normal aging and early Alzheimer's disease, early cognitive deficits are often

apparent [3–6]. A significant number of patients with MCI will eventually develop AD within a few years once they develop the disease [7].

The ability to learn and memorize is essential for our survival since it allows us to adapt our behavior and enhances our ability to survive [1]. It is well established that neurobehavioral test results, long-term cognitive decline, and dementia risk differ significantly among races/ethnicities, with African Americans scoring lower than whites [7–11].

A standard periodic lexical recognition test is used in clinics for assessing elderly cognitive performance called the Rey Auditory Verbal Learning Test (RAVLT) [12]. Participants take the RAVLT to assess their ability to encode, consolidate, store, and retrieve verbal information [1,13,14]. MCI to AD progression is predicted by the RAVLT learning score (V-I) [15–19].

The neuropsychological performance of non-Hispanic African American and non-demented white older adults was also measured in Northern Manhattan. Different parameters such as verbal/nonverbal learning, memory, and abstract reasoning revealed a significant difference between African American participants and their white counterparts [20].

Education is one of the potential factors that may decrease the onset of cognitive decline. The cognitive performance levels of highly educated people are superior in virtually every domain of cognition. In contrast, a low level of education is associated with an increased risk of Alzheimer's disease [2–5,21].

A person's education level is commonly used as an indicator of their cognitive reserve [4,6,8]. This notion is validated by the correlation between a higher education degree and a lower risk of dementia. The results of prospective studies suggest that the relationship is primarily driven by education and cognitive function, as opposed to the function of the brain alone [9–13].

Cognitive reserve and education are controversial concepts. Studies, such as the Religious Orders Study/Memory and Aging Project (ROSMAP) [22] and Bronx Aging Study, have emphasized the crucial role of a higher education degree in managing cognitive decline. Contrary to this, other studies contradict it [23–25]. Additionally, the Religious Orders Study revealed that higher education was associated with a lower negative association between AD pathology and cognition-related death [26].

Kungsholmen Project, a longitudinal study, examined the impact of education on dementia using sex, age, and educational data [27,28]. Participants in this study were divided into three categories based on their years of schooling: 8 years, 8–10 years, and 11 years in college. The findings of this study suggest that low-educated participants are more likely to develop dementia or Alzheimer's disease.

People who are married have more frequent social communication, which reduces the risk of dementia. Individuals with neuropathological impairments are capable of maintaining their cognitive skills and daily activities despite the impairment [29,30]. Conversely, divorced people always experience stress that increases their risk of dementia [31,32]. Moreover, the ages, races, and sexes of married people are also factors related to cognitive impairment [33–35].

## 2. Materials and Methods

### 2.1. The TADPOLE Data Study Participants

As part of a collaboration with the Alzheimer's Disease Neuroimaging Initiative (ADNI), we selected data that were collected from the Brain Damage Prediction of Longitudinal Evolution (TADPOLE) to test the hypothesis [36]. A list of participants recruited to the Alzheimer's Disease Neuroimaging Initiative (ADNI) is available on TADPOLE. Some of these individuals have already contributed data to earlier ADNI studies and have agreed to provide follow-up information. In TADPOLE, each feature corresponds to a likely clinical trial outcome: CN—Cognitively normal; MCI—mild cognitive impairment;

or AD—Alzheimer's disease. The TADPOLE database contains a number of biological markers, some of which are missing data, but are still very informative.

Crucial biomarkers with ADNI-TADPOLE are:

- Target outcomes: ADAS13, DX, Ventricles;
- Cognitive tests: MMSE, ADAS11, CDRSB, RAVLT-immediate;
- MRI measures: WholeBrain, Hippocampus, Entorhinal, MidTemp;
- PET measures: AV45, FDG;
- CSF measures: (amyloid-beta level in CSF), TAU (tau level), (phosphorylated tau level);
- Risk factors: AGE, APOE4.

We used our previously published method [37] to target cognitive tests that are confirmed to be valid. In order to test our hypothesis, we decided to use RAVLT as a complete exposure variable instead of EcogPt.

### 2.2. Outcome: Normal Cognitive vs. MCI

The primary outcome is whether participants have typical cognition or mild cognitive impairment [36]. Mild cognitive impairment (MCI) marks the transition from normal cognitive aging to progressive cognitive decline [38]. The dataset contains a group that reverses the disease from MCI to normal cognitive function [36]. Transitional cognitive decline is pervasive in Alzheimer's disease (AD) participants. Therefore, cognitive decline can be based on a subjective report by the individual or participant in the MCI group [39].

TADPOLE is the result of combining various ADNI phases into one dataset. During the first phase of ADNI-1, data were collected in 2004. There were 821 participants with MCI in total. There are two additional phases, ADNI-GO and ADNI-2, with 200 early MCI participants. There are currently up to 1200 additional participants registered in ADNI-3, which is enrolling participants with average cognitive ability (AC), mild cognitive impairment (MCI), and Alzheimer's disease (AD) [40].

### 2.3. Rey Auditory Verbal Learning Test (RAVLT)

Rey Auditory Verbal Learning Test (RAVLT) measures episodic memory. Cognitive decline caused by abnormal aging can be detected with RAVLT [39]. In previous ADNI work, we found RAVLT to be one of the most influential measures in the prediction model (Ensemble Learning Model with Feature Selection), which explains our interest in it.

### 2.4. Performance for RAVLT

Research has shown that Rey's auditory verbal learning test (RAVLT) can be used to detect Alzheimer's disease early [16]. RAVLT is also significantly influential in differentiating AD from psychiatric disorders [41–43]. Our study focused on participants with MCI. In previous research, no corresponding association was found between MCI that occurred early or late. False-positive diagnostic errors also occur in a large percentage [44].

A RAVLT estimates a total of four mini-tests: learning rate, short-term auditory-verbal memory learning strategies, retroactive interference, and proactive interference. The RAVLT test evaluates confusion in memory processes, retention of information, and learning–retrieval differences.

According to our hypothesis, RAVLT performance with a marital status of never married may be associated with greater odds of MCI than either RAVLT performance on its own. A study of the RAVLT showed that intelligent and gifted students scored higher than typical students [16], despite studying intelligence more holistically and comprehensively than just memorizing facts. This evaluation of short-, working-, and long-term memory is vital to providing basic daily services to older adults participating in ADNI.

The participants repeat 15 unrelated words over five trials in TADPOLE. They must repeat the original 15 words after 30 min, followed by 15 unrelated words. The procedure takes approximately 10 to 15 min (excluding intervals of 30 min) [17].

Our study with logistic regression assesses four RAVLT subgroups: low- and high-performance in the test for an immediate response. We add the other three subgroups: learning, forgetting, and perception of forgetting to the final model.

We constructed a RAVLT-modified model with low- and high-performance adjusted for age, sex, race, engagement, ethnicity, education, APOE4 genotype, hippocampus, whole brain, ventricles, and ICV.

### 2.5. Main Confounders

We adjusted for the following sociodemographic covariates: age, gender, race, ethnicity, APOE4 genotype (0, 1, or 2 alleles), race (White, Black or African American, Asian, American Indian or Alaska Native or Native Hawaiian or Other Pacific Islander), engagement (Married, Never Married, Unknown, Widowed), education (Elementary, High School, Occupational Program, College, Post-College), hippocampus (Volumetric measurements), ethnicity (not Hispanic or Latino, Hispanic or Latino, or unknown), whole-brain (Volumetric measurements), ventricles (Volumetric measurements), and ICV (Volumetric measurements).

### 2.6. Analytical Sample

Our analytical sample from TADPOLE longitudinal analysis used baseline follow-up visit data from three "standard" data sets, derived from ADNI-1, ADNI-GO, and ADNI-2.

Standard data sets are:

- D1—a comprehensive longitudinal data set for training;
- D2—a comprehensive longitudinal data set on rollover subjects for forecasting;
- D3—a limited forecasting data set on the same rollover subjects as D2.

The initial analytical sample considered all participants with baseline cognitive and educational data, we have 12,741 participants from standard datasets. Participants with dementia and reverse MCI to Normal were excluded (6141), leaving 6600 participants in total. The final analytical sample consisted of 6560 participants (Figure 1).

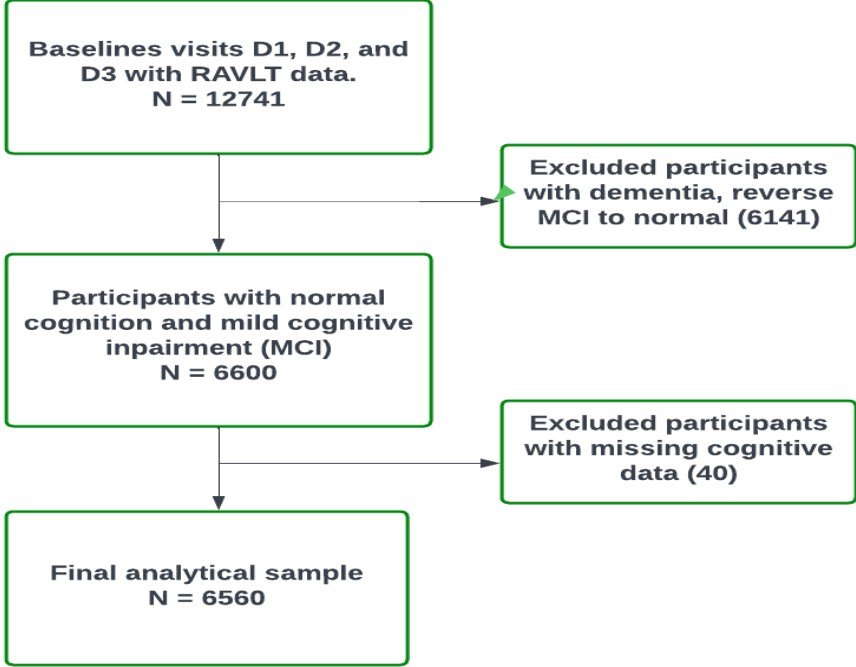

**Figure 1.** Diagram of analytical sample.



### 2.7. Descriptive Analysis of Significant Exposure Variables

RAVLT consists of five repeated learning trials of a fifteen-word list, followed by immediate and delayed recall trials after three and thirty minutes, respectively, and recognition trials after each. Memory deficits in AD have been examined in much of the literature by either summing words recalled across all learning trials or using delayed recall measures [45], or sometimes by considering list learning performance as a composite measure of episodic memory.

Figures 2 and 3 show the RAVLT test of immediate response, learning test, forgetting, and perception of forgetting as a sum of five trials based on education level. The distribution of participants with unknown marital status was relatively concentrated.

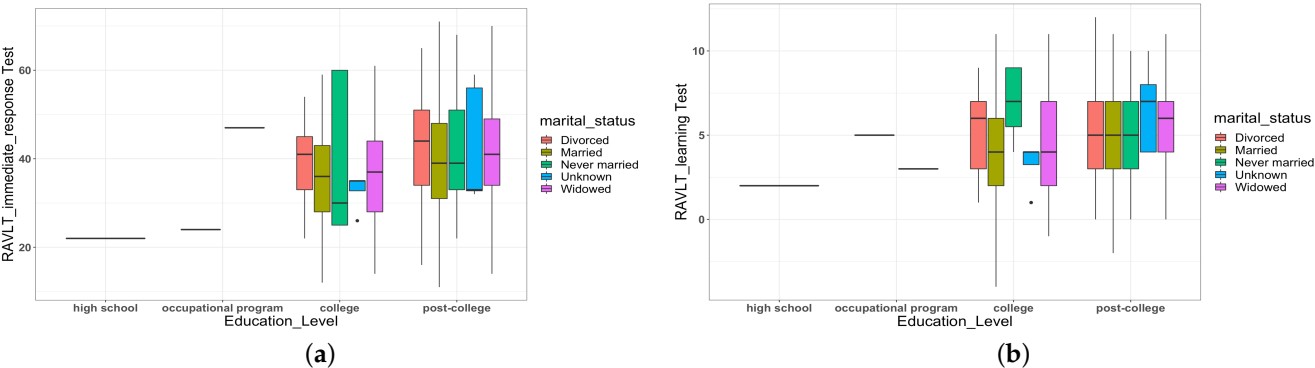

**Figure 2.** Relatively concentrated distributions of unknown: marital status with college education vs. RAVLT test of immediate response and learning. (**a**) RAVLT immediate response test vs. marital status and education; (**b**) RAVLT Learning test vs. marital status and education.

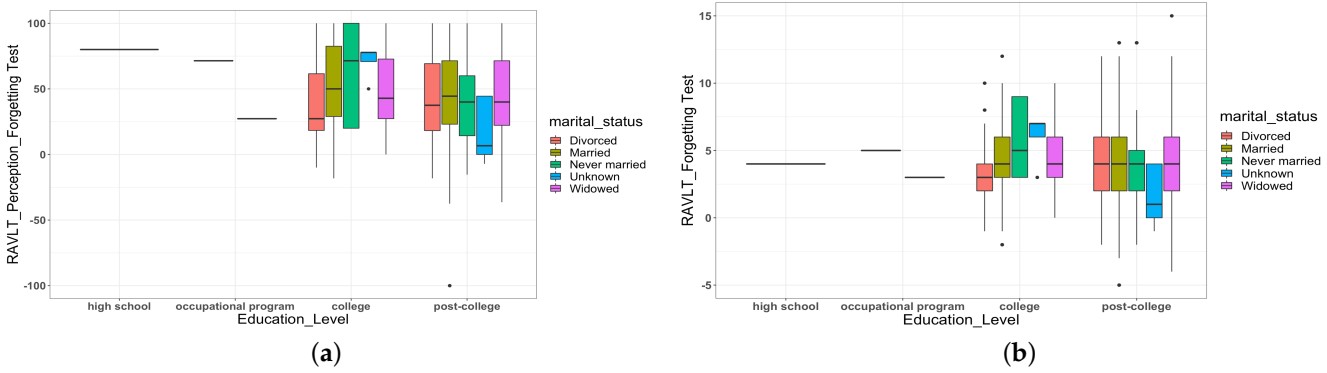

**Figure 3.** Relatively concentrated distributions of unknown: marital status with college education vs. RAVLT test of forgetting and perception of forgetting. (**a**) RAVLT perception of forgetting test vs. marital status and education; (**b**) RAVLT forgetting test vs. marital status and education.

### 3. Results

In Table 1, we present descriptive statistics for the sample among the MCI and NL groups. Participants in the sample were more likely to be 71+, with 58 percent being MCI. The study group possessed 62% APOE4 alleles, 93% whites, 97% non-Hispanics, and 62% MCI.

**Table 1.** Demographic characteristics of the study population per RAVLT test subgroup.

| Category | Level | MCI Group | NL Group | | |
|---|---|---|---|---|---|
| | | Total (n, %) | MCI Group (n, %) | NL Group (n, %) | Chi-Squared *p*-Value |
| Auditory Verbal Learning Test Performance (RAVLT) | Immediate Response | 39 (31, 48) | 34 (28, 42) | 45 (38, 52) | $2.00 \times 10^{-16}$ |
| | Learning | 5 (3, 7) | 4 (2, 6) | 6 (4, 8) | $2.35 \times 10^{-06}$ |
| | Forgetting | 4 (2, 6) | 5 (3, 6) | 4 (2, 5) | $2.00 \times 10^{-06}$ |
| | Perception of forgetting | 44 (23, 72) | 57 (31, 86) | 31 (15, 50) | $2.00 \times 10^{-06}$ |
| Sex | Male | 3690 (56) | 2373 (60) | 1317 (49) | 0.11226 |
| | Female | 2910 (44) | 1559 (40) | 1351 (51) | |
| Age | Age 0 to 60 | 195 (3.6) | 167 (5.3) | 28 (1.2) | |
| | Age 61 to 70 | 1652 (31) | 1144 (36) | 508 (23) | $7.04 \times 10^{-07}$ |
| | Age 71 or older | 3542 (66) | 1827 (58) | 1715 (76) | $2.00 \times 10^{-16}$ |
| Race | White | 6128 (93) | 3685 (94) | 2443 (92) | 0.01202 |
| | Black/ African American | 271 (4.1) | 116 (3.0) | 155 (5.8) | 0.003855 |
| | Hawaiian | 5 (<0.1) | 5 (0.1) | 0 | 0.976919 |
| | Asian | 114 (1.7) | 73 (1.9) | 41 (1.5) | 0.126461 |
| | Unknown | 12 (0.2) | 12 (0.3) | 0 | 0.944922 |
| | More than one | 56 (0.8) | 32 (0.8) | 24 (0.9) | 0.000166 |
| Ethnicity | Hispanic/Latino | 199 (3.0) | 124 (3.2) | 75 (2.8) | 0.756096 |
| | Not Hispanic/Latino | 6370 (97) | 3794 (96) | 2576 (97) | |
| | Unknown | 31 (0.5) | 14 (0.4) | 17 (0.6) | 0.023799 |
| Marital Status | Married | 4914 (74) | 3016 (77) | 1898 (71) | 0.887499 |
| | Never Married | 222 (3.4) | 81 (2.1) | 141 (5.3) | $3.10 \times 10^{-10}$ |
| | Unknown | 36 (0.5) | 31 (0.8) | 5 (0.2) | 0.007498 |
| | Widowed | 824 (12) | 446 (11) | 378 (14) | 0.816086 |
| Education | Elementary | 0 | 0 | 0 | |
| | High School | 2 (<0.1) | 2 (<0.1) | 0 | |
| | Occupational Program | 11 (0.2) | 1 (<0.1) | 10 (0.4) | 0.981215 |
| | College | 1240 (19) | 859 (22) | 381 (14) | 0.983097 |
| | Post-Collage | 5347 (81) | 3070 (78) | 2277 (85) | 0.983097 |
| APOE4 alleles | 0 | 4077 (62) | 2130 (54) | 1947 (73) | |
| | 1 | 2102 (32) | 1429 (36) | 673 (25) | |
| | 2 | 417 (6.3) | 370 (9.4) | 47 (1.8) | |
| Hipocampus | Hipocampus | 7177 (6383, 7817) | 6958 (6129, 7687) | 7431 (6787, 7918) | $2.00 \times 10^{-16}$ |
| Wholebrain | Wholebrain | 1,033,900 (961,397, 1,104,800) | 1,034,980 (962,600, 1,113,680) | 1,033,180 (959,755, 1,096,880) | $3.35 \times 10^{-09}$ |
| Ventricles | Ventricles | 32,311 (21,694, 46,110) | 33,924 (23,127, 49,959) | 29,415 (20,203, 40,820) | 0.054315 |
| ICV | ICV | 1524520 (1,422,670, 1,640,600) | 1534150 (1,436,660, 1,647,330) | 1505460 (1,401,292, 1,630,490) | 0.045143 |

Some significant differences existed between the four stratified RAVLT tests (Table 1). Participants at risk of MCI were more likely to be male, regardless of cognitive test status.

Only immediate response tests (OR 0.96, 95% CI 0.95–0.97) and forgetting tests (OR 0.65, 95% CI 0.60-0.69) were positively associated with MCI odds in logistic regression models of RAVLT subgroups and MCI (Table 2). The learning subgroup test was not significantly associated with elevated odds of MCI relative to RAVLT alone. As a result of logistic regression models with model 2 adjusted for age and gender as a reference, immediate response (OR 0.95, 95% CI 0.94–0.96), forgetting (OR 0.59, 95% CI 0.55–0.65) and age 61 to 70 (OR 0.34, 95% CI 0.22–0.53) and older (OR 0.09, 95% CI 0.06–0.15) were all negatively associated with MCI (Table 2).

Finally, model 3 was adjusted for sex, race, ethnicity, marital status, education, APOE4, hippocampus, whole-brain, ventricles, and ICV as the reference, only immediate response (OR 0.96, 95% CI 0.95–0.97) and forgetting (OR 0.65, 95% CI 0.60–0.69) age 61 to 70(OR 0.26, 95% CI 0.15–0.45), age 71 or older (OR 0.07, 95% CI 0.04–0.12), Race-Black/African American (OR 0.13, 95% CI 0.03–0.52), more than one (OR 0.05, 95% CI 0.01–0.24), and marital status—never married (OR 0.2, 95% CI 0.12–0.34) were each negatively associated with increased odds of MCI (Table 2). In Appendix A, we describe indications where results can be reproduced by our own curated data.

**Table 2.** Associations between RAVLT test for learning, forgetting, immediate response, and perception of forgetting and MCI with low and high risk for RAVLT test alone or fully adjusted. Covariates with symbol ** are most negatively associated covariates compared with only one symbol *.

| Category | MCI Group Level | Model 1: Unadjusted Odds Ratio | 95% Confident Interval | Model 2: Adjusted for Age and Sex Odds Ratio | 95% Confident Interval | Model 3: Fully Adjusted Odds Ratio | 95% Confident Interval |
|---|---|---|---|---|---|---|---|
| Auditory Verbal Learning Test Performance (RAVLT) | No Association | Reference | Reference | Reference | Reference | Reference | Reference |
| | Immediate response | 0.96 | 0.95–0.97 | 0.95 | 0.94–0.96 | 0.94 | 0.93–0.95 * |
| | Learning | Reference | Reference | Reference | Reference | 1.1 | 1.05–1.14 ** |
| | Forgetting | 0.65 | 0.60–0.69 | 0.59 | 0.55–0.65 | 0.57 | 0.51–0.60 * |
| | Perception of forgetting | 1.05 | 1.04–1.05 | 1.06 | 1.05–1.07 | 1.06 | 1.05–1.07 ** |
| Sex | Male | NA | NA | Reference | Reference | Reference | Reference |
| | Female | NA | NA | Reference | Reference | Reference | Reference |
| Age | Age 0 to 60 | NA | NA | Reference | Reference | Reference | Reference |
| | Age 61 to 70 | NA | NA | 0.34 | 0.22–0.53 | 0.26 | 0.15–0.45 * |
| | Age 71 or older | NA | NA | 0.09 | 0.06–0.15 | 0.07 | 0.04–0.12 ** |
| Race | White | NA | NA | NA | Reference | Reference | Reference |
| | Black/African American | NA | NA | NA | NA | 0.13 | 0.03–0.52 ** |
| | Asian | NA | NA | NA | NA | Reference | Reference |
| | More than one | NA | NA | NA | NA | 0.05 | 0.01–0.24 ** |
| Ethnicity | Hispanic/Latino | NA | NA | NA | NA | Reference | Reference |
| | Not Hispanic/Latino | NA | NA | NA | NA | Reference | Reference |
| | Unknown | NA | NA | NA | NA | Reference | Reference |
| Marital Status | Married | NA | NA | NA | NA | Reference | Reference |
| | Never Married | NA | NA | NA | NA | 0.2 | 0.12–0.34 ** |
| | Unknown | NA | NA | NA | NA | 5.01 | 1.53–0.16 |
| | Widowed | NA | NA | NA | NA | Reference | Reference |
| Education | Elementary | NA | NA | NA | NA | Reference | Reference |
| | High School | NA | NA | NA | NA | Reference | Reference |
| | Occupational Program | NA | NA | NA | NA | Reference | Reference |
| | College | NA | NA | NA | NA | Reference | Reference |
| | Post-Collage | NA | NA | NA | NA | Reference | Reference |
| APOE4 alleles | 0 | NA | NA | NA | NA | 1.81 | 1.58–2.08 |
| | 1 | NA | NA | NA | NA | Reference | Reference |
| | 2 | NA | NA | NA | NA | Reference | Reference |
| Hippocampus | Hippocampus | NA | NA | NA | NA | 0.99 | 0.99–0.99 |
| Whole-brain | Whole-brain | NA | NA | NA | NA | 1 | 1.00–1.00 |
| Ventricles | Ventricles | NA | NA | NA | NA | Reference | Reference |
| ICV | ICV | NA | NA | NA | NA | Reference | Reference |

## 4. Discussion

We assessed cross-sectional associations between RAVLT test performance in immediate response, learning, forgetting, and perception of forgetting and marital status levels and MCI using Alzheimer's Disease Prediction Of Longitudinal Evolution (TADPOLE) data. TADPOLE identifies various approaches that are crucial prognostic indicators of future AD progression. With TADPOLE, a significant set of multimodal assessments from the Alzheimer's Disease Neuroimaging Initiative (ADNI) is utilized for predicting disease progression [46].

A very limited number of rigorous studies have examined the association between the RAVLT cognitive tests and the marital status levels of the MCI group using the TADPOLE dataset. The purpose of this study is, therefore, to investigate the association between RAVLT test performance and marital status and MCI through the use of the TADPOLE data set.

A previous study indicated a substantial correlation between information provided by RAVLT scores and AD-linked structural atrophy. Indeed, participants from similar groups such as "AD and MCI" or "Normal control and MCI" produced lower predictive performance compared to other groups of subjects with significant structural changes in the brain, such as "AD and Normal Control" [47]. Furthermore, another study on people with prodromal AD (amyloid- and tau-positive amnestic mild cognitive impairment) realized that episodic memory impairment is also triggered by neurodegeneration in numerous cortical networks outside of the standard memory systems [48].

Atrophies in AD are not random; rather, they follow a well-defined path that starts with the entorhinal cortex and the hippocampus and eventually affects all cortex regions. Individuals with MCI had significantly higher rates of annual atrophy than those with normal aging. The rate of atrophy was continuously increasing, accompanied by an increase in AD [49]. In MCI patients, there is an association between p-tau levels and temporal atrophy, whereas it is not observed in healthy elderly people [49].

The RAVLT score is a crucial tool for predicting the progression of MCI into AD [9–13]. Model 2 was negatively associated with MCI when age 61 to 70, >71 and sex were adjusted, as were the immediate response and forgetting behaviors. Research on the cognitive assessment of cognitively normal individuals revealed that demographic parameters, particularly age, have a substantial impact on the RAVLT assessment of memory [50]. When normal aging influences were excluded from the measured RAVLT scores, there was a small improvement. This observation was linked to the impact of AD pathology on MRI and RAVLT, dominating the effects of normal aging [47].

MCI was negatively associated with two subgroups of RAVLT (immediate response and forgetting) in our logistic regression model for fully adjusted RAVLT. The final model revealed no significant association between the learning subgroup and the perception of forgetting test.

Our analyses also agree with those previously highlighting the role of education level and socio-economic status in maintaining cognitive function and protecting against AD diagnosis. The community-based longitudinal study, Kungsholmen Project, highlighted the influence of education on dementia incidence using data of demographic variables such as sex, age, and educational level [28]. Participants of this study were assigned to three categories of schooling < 8 years of school, 8–10 years, and the university for those with 11 years of education. Data analysis for this study suggests that participants with a low level of education are at high risk of developing dementia or AD, especially women.

In parallel with our analyses, many studies have demonstrated that longitudinal brain changes (as measured by MRI) are linked to changes in cognitive capacities, particularly episodic memory tasks [9–13]. Amyloid deposition is considered a major biomarker of the histopathological classification of AD [51]. In neurodegenerative disease, amyloid is projected as the lead of detrimental events that progress to dementia and AD [52]. Neuroplasticity is a well-known cognitive decline that evolves when the brain fails to compensate for accumulating insults. Healthy normal aging and the earliest stages of AD may be related to neuroplasticity [31,45].

Various studies outside and inside the United States highlighted the marital status difference as a crucial lead in cognitive impairment and dementia. Studies conducted outside the United States declared that Alzheimer's was substantially more likely to affect single men and women than married people, and widowed [53–56]. In contrast, widowhood is considered a risk factor for being diagnosed with Alzheimer's among American old residents [57]; however, men were more vulnerable than women [33–35]. Furthermore, a recent study involving an old population of 65 or more years old confirmed these previous studies and found marital status as a protective aspect for cognitive impairment, but those who were previously married (widowed and divorced) were at high risk of cognitive impairment [58].

The results cited previously are in line with our findings, but as was expected in many studies, there are some limitations, which compel further investigation. Our current models should be validated by considering more variables in future research.

## 5. Limitations

This study has limitations inherent to cross-sectional studies like not being able to address temporality. We measure for two exposure variables: RAVLT and marital status, but in cross-sectional studies exposures, variables do not precede certain outcomes. Our use of RAVLT test does not avoid the complexity of having false positive subjects, who have the presence of brain dysfunction, errors occur when intact individuals are labeled as

having brain dysfunction. The TADPOLE dataset is not a population-based sample. We can have by design, people with MCI and dementia over-sampled. We may infer that the results may not generalize to the community-dwelling population.

## 6. Conclusions

This design research may suggest those older individuals that never married, and performed poorly in RAVLT test of immediate response and forgetting may be a group at high risk for mild cognitive impairment. There are significant data on social engagement that point to a protective effect of higher levels of middle and late-life social engagement, reducing the risk of Alzheimer's disease and related dementias (ADRDs). Our hypothesis is that RAVLT cognition performance with marital status is associated with greater odds of MCI group than RAVLT independently. To this end, we used TADPOLE data to evaluate cross-sectional associations between RAVLT performance in immediate response, learning, forgetting, and perception of forgetting with marital status and MCI.

Our methods involved subjects with MCI and normal cognition. Logistic regression models indicate strong associations among four RAVLT subgroups ((1) low and high performance of immediate response, (2) immediate response with learning, (3) performance of immediate response with learning and forgetting, (4) performance of immediate response with learning, forgetting, and perception of forgetting) and MCI group. Models were adjusted for age, sex, race, marital status, ethnicity, education, APOE4 genotype, Hippocampus, whole brain, ventricles, and ICV. TADPOLE underlines various approaches that are crucially prognostic of the future progression of people at risk of AD. TADPOLE utilizes a substantial set of multimodal assessments from ADNI, which endorses the prediction of the disease progression.

Studies are needed to evaluate another cognitive test with missing data within the TADPOLE dataset as a modifiable risk factor for MCI.

**Author Contributions:** Conceptualization, Y.J.P.B., R.A. and C.Y.; methodology, Y.J.P.B., C.T., R.A. and S.S.M.; formal analysis, Y.J.P.B., R.A. and S.S.M.; data curation, Y.J.P.B., S.B. and V.A.; writing—original draft preparation, Y.J.P.B. and S.S.M.; writing—review and editing, Y.J.P.B., S.S.M. and R.A.; visualization, Y.J.P.B., C.T., S.S.M. and V.A.; supervision, Y.J.P.B. and R.A. All authors have read and agreed to be published version of the manuscript.

**Funding:** This research was supported by National Health Institute Grant in Biomedical applications. Number #006580.

**Institutional Review Board Statement:** The study was conducted according to the guidelines of the declaration of Helsinki and approved by the local institutions review boards of each participant ADNI site. A complete list of participating ADNI sites and IRB details can be obtained at adni.loni.usc.edu accessed on 15 January 2023.

**Informed Consent Statement:** Informed consent was obtained from all subjects or authorized representative involved in the study. Details of informed consent procedures can be obtained at adni.loni.usc.edu accessed on 15 January 2023.

**Data Availability Statement:** The data used in this study are available from the Alzheimer's Disease Neuroimaging Initiative Study Database (adni.Lni.usc.edu accessed on 15 January 2023).

**Acknowledgments:** Institutional support was provided by the college of science and technology at Florida A&M University. Data collection and sharing for this project were founded by the Alzheimer's Disease Neuroimaging Initiative (ADNI) though the TADPOLE challenge. ADNI is funded by the National Institute of Aging, the National Institute of Biomedical Imaging anf Bio-engineering, and through generous contributions from the following: AbbVie; Alzheimer's Association; Alzheimer's Drug Discovery Association; Araclon Biotech; BioClinica, Inc.; Biogen; Bristol-Myers Squibb Company; CereSpir, Inc.; Cogstate, Eisai Inc.; Elan Pharmaceuticals, Inc.; Eli Lilly and Company; EuroImmun; F. Hoffman-La Roche Ltd. and its affiliated company Genentech, Inc.; Fujirebio; GE Healthcare; IXICO Ltda.; Janssen Alzheimer Immunotherapy Research & Development, LLC.; Johnson & Johnson Pharmaceutical Research & Development LLC.; Lumosity; Lundbeck; Merck & Co., Inc.; Meso Scale Diagnostics, LLC.; NeuroRx Research; Neurotrack Technologies; Novartis Pharmaceuticals

Corporation; Pfizer Inc; Piramal Imaging; Servier; Takeda Pharmaceutical Company; and Transitions Therapeutics. Private sector contributions are facilitated by the foundation for the National Institutes of Health (www.fnih.org accessed on 15 January 2023) The grantee organization is the Northern California Institute for Research and Education, and the study is coordinated by the Alzheimer's Therapeutic Research Institute at the University of Southern California. ADNI data are disseminated by the laboratory for Neuro Imaging at the University of Southern California.

**Conflicts of Interest:** The authors declare no conflict of interest. The funders had no role in the design of the study; in the collection, analysis, or interpretation of data; in the writing of the manuscript; or in the decision to publish the results.

## Abbreviations

The following abbreviations are used in this manuscript:

| | |
|---|---|
| RAVLT | Rey Auditorial Verbal Learning Test |
| MCI | Mild Cognitive Impairment |
| ADNI | Alzheimer's Disease Neuroimaging Initiative |
| TADPOLE | Alzheimer's Disease Prediction of Longitudinal Evolution |

## Appendix A

*Appendix A.1*

The analysis done using an R programming language. The code and dataset are available on Kaggle.

You may follow the following steps to see and/or reproduce the results:

1. Click on the Kaggle link (https://www.kaggle.com/code/victorwealth/covariates-association-tadpole accessed on 15 January 2023).
2. Click on Register and complete the sign-up process to create a Kaggle account or Login if you already possess an account.
3. Once logged in, click on Copy & Edit to make a copy of the code for your personal use.
4. In the copied version, you can either run per cell or run all cells to see the results.

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
