# Peer review of "Marital Status of Never Married with Rey Auditory Verbal Learning Test Cognition Performance Is Associated with Mild Cognitive Impairment"

_applsci, doi:10.3390/app13031656_

Round 1

Reviewer 1 Report

1.) In the abstract the goal is decribed as: "To test the hypothesis that RAVLT cognition performance with marital status is associated with greater odds of MCI group than either RAVLT independetly". This is also statd in the discussion "to evaluate RAVLT performance with marital status and MCI". However correction in the final model 3 is done with multiple variables from which marital status-never is one of them. While it is appreciated that not only the variable "married" was used, it illustrate that it is completely unclear what the goal of the study is. 

2.) Demographic characterisics for the marital status is not given. The Odds ration of unknown is very high. Is it true that for most of the participants the marital status is not known. Thus that the primary question can't be addressed. 

3) 93% of participants are white. You can not make any statement about ethnicity, as all other ethnees are completely unterrepresentated.

4) Figure 2 a has to be shown as dot blots. Again that would illustrate better the number of participant per group. Again showing other races than white is not really informativ as it is compeltely underpowered. 

5) Figure 2 c. What do you want to say with that? What is the x axis telling us (5'000, 6'000) etc? Why is this result not explained in the text? 

6) Discussion is poorly (if any) linked to the result. For example what do you want to say with the paragraph between 247-254?

Reviewer 2 Report

Major comments:

The authors need to improve the flow of this manuscript. It is unclear why the authors emphasize marital status in the title and abstract using the phrase “Marital Status with Rey Auditory Verbal Learning Test” or “Rey Auditory Verbal Learning test (RAVLT) cognition performance with marital status”. Because the results, discussion, and in particular, conclusions sections also discussed and focused on other significant covariates, the authors should extensively edit the whole manuscript and put it in the context of why marital status needs to be highlighted. 

Minor comments:

Abstract: 

Please spell out ADNI in the first use.

Introduction: 

The authors did not articulate the current knowledge gap and the aims of this study.

Methods:

“Figure 2 shows RAVLT immediate response test as a sum of five trails. Participants that never married in black race shows a relatively concentrated distributions. However, significant relationship between both with respect RAVLT as we show in table 2.”

This sentence reads like results, instead of methods.

Results:

“At the same time, positive association compared to those who have RAVLT test alone, those with RAVLT learning test, being black significant hippocampus measurements and never married had an average 1% higher odds of being MCI, although this relationship was not statistically significant (Table 2).” This sentence is grammatically incorrect and should be rewritten.

Reviewer 3 Report

In this paper, the authors used TADPOLE data to evaluate cross-sectional associations between RAVL performance in immediate response, learning, forgetting, and perception of forgetting with marital status and MCI. This manuscript is worthy of publication, but the authors need to resolve the following suggestions before publication:

1.     The introduction is not well arranged, and the logic is not clear.

2.     The conclusion did not mention the correlation between marital status with RAVLT and mild cognitive impairment.

3.     There are many grammatical errors throughout the manuscript and need to be carefully revised.

Round 2

Reviewer 1 Report

The authors improved the paper. It can be accepted now. 

Author Response

Thank you. 

Reviewer 2 Report

The authors did not address/respond to my major comments. And I think the authors still need to improve the writing of this manuscript.

Author Response

We address again and respond to the significant comments:

  1. Race is not an outcome variable in our logistic regression study. We change graphs 2 and 3 to convey the message.
  2. We found positive and negative associations, which are now explained in the paper. 
  3. The title was modified to convey the message.